# A novel ALS-associated variant in *UBQLN4* regulates motor axon morphogenesis

Brittany M Edens[1], Jianhua Yan[2], Nimrod Miller[1], Han-Xiang Deng[2], Teepu Siddique[2], Yongchao C Ma[1]*

[1]Department of Pediatrics, Northwestern University Feinberg School of Medicine, Ann & Robert H Lurie Children's Hospital of Chicago, Chicago, United States; [2]The Ken & Ruth Davee Department of Neurology, The Les Turner ALS Research and Patient Center, Northwestern University Feinberg School of Medicine, Chicago, United States

**Abstract** The etiological underpinnings of amyotrophic lateral sclerosis (ALS) are complex and incompletely understood, although contributions to pathogenesis by regulators of proteolytic pathways have become increasingly apparent. Here, we present a novel variant in *UBQLN4* that is associated with ALS and show that its expression compromises motor axon morphogenesis in mouse motor neurons and in zebrafish. We further demonstrate that the ALS-associated *UBQLN4* variant impairs proteasomal function, and identify the Wnt signaling pathway effector beta-catenin as a *UBQLN4* substrate. Inhibition of beta-catenin function rescues the *UBQLN4* variant-induced motor axon phenotypes. These findings provide a strong link between the regulation of axonal morphogenesis and a new ALS-associated gene variant mediated by protein degradation pathways.

*For correspondence: ma@northwestern.edu

Competing interests: The authors declare that no competing interests exist.

## Introduction

Ubiquilins belong to the ubiquitin-like family of proteins and act broadly as key regulators of degradative processes. The best understood function of ubiquilins is in the ubiquitin proteasome system (UPS), where they recognize and bind polyubiquitinated substrate proteins through a C-terminal ubiquitin-associated (UBA) domain and deliver them to the proteasome through an N-terminal ubiquitin-like (UBL) domain (*Marín, 2014*). Additional roles for ubiquilins in autophagy and ER-associated degradation have also been described (*Hjerpe et al., 2016*; *Lee and Brown, 2012*; *Lim et al., 2009*). The UPS and autophagy are indispensable for multiple aspects of neuronal development (*Cecconi and Levine, 2008*; *Hamilton and Zito, 2013*) and are crucial in maintaining homeostasis in the aging nervous system (*Groen and Gillingwater, 2015*; *Rezania and Roos, 2013*). Dysfunction of these pathways contributes to neurodegenerative diseases characterized by protein aggregation (*Huang et al., 2010*; *Kim et al., 2009*; *Wang et al., 2011*; *Wu et al., 2015*). Accordingly, variants in *UBQLN1* and *UBQLN2* have been linked to neurological disorders. The ubiquilin1 protein, encoded by *UBQLN1*, has been associated with neurofibrillary tangles in Alzheimer's disease (AD) and is proposed to contribute to protein aggregates characteristic of AD pathology (*El Ayadi et al., 2012*; *Mah et al., 2000*). Single-nucleotide polymorphisms in *UBQLN1* were suggested to confer susceptibility to Alzheimer's disease (*Bertram et al., 2005*; *Kamboh et al., 2006*). Mutations in *UBQLN2* have been related to familial X-linked ALS/FTD (*Deng et al., 2011*) and heterogeneous X-linked dominant neurodegeneration (*Fahed et al., 2014*). Here, we report a novel variant in *UBQLN4* that is associated with ALS and demonstrate a mechanism by which wild-type *UBQLN4* may regulate

**eLife digest** Amyotrophic lateral sclerosis, or ALS for short, is a disease in which parts of the brain and spinal cord progressively degenerate. Specifically, the condition causes the nerve cells that control movement – called motor neurons – to die. As a result, people with ALS lose control of their muscles. The cause of ALS is not known, but evidence suggests that a person's genetics plays a role in the development of the disease. Learning which genes are involved and what they do within cells may help scientists figure out what goes wrong in patients with ALS and how to treat the condition.

People with ALS often experience an abnormal build up of proteins in their brain and spinal cord. Cells normally rely on molecules working together in the so-called ubiquitin proteasome system to eliminate unwanted proteins. A few mutations linked with ALS, and some other neurodegenerative conditions, have been traced back to genes encoding parts of this protein disposal system, which may help to explain the build up of proteins. However, our understanding of the genetic causes of the disease is far from complete.

Now, Edens et al. report a new mutation in a gene that encodes a protein involved in the ubiquitin proteasome system. The gene, called *UBQLN4*, had not previously been linked to ALS, but looking at this gene in nearly 700 patients with ALS revealed a mutation in one patient with an inherited form of the disease. This mutation was not found in public databases that contain genetic information from tens of thousands of people without ALS.

To better understand the effect of this newly identified mutation, Edens et al. recreated it in zebrafish embryos and motor neurons from mice. The mutation in *UBQLN4* changed the shape of the cells in the spinal cords of the zebrafish and the mouse motor neurons. There was also a build up of excess proteins because the breakdown of proteins by the ubiquitin proteasome system was slowed. Specifically, there was an excess amount of a protein called beta-catenin, which is important for development and activity of the nervous system. Treating the mutant motor neuron cells with a drug called quercetin, which suppresses beta-catenin, reversed the defects seen in the cells. Larger studies are now needed to see how often this mutation occurs in patients with ALS and to determine if other forms of the disease might have a similar cause.

motor axon morphogenesis. We also reveal how dysregulation of this mechanism by the ALS-associated variant leads to abnormal motor neuron structure and function through impaired degradation and substrate retention.

## Results

### Identification of an ALS-associated *UBQLN4* variant in familial ALS that affects motor axon morphogenesis

Mutations in *UBQLN2* have been identified in ALS patients with or without dementia (**Deng et al., 2011**). A variant, E54D, in *UBQLN1* has been reported in a single patient with atypical motor neuron disease consistent with Brown-Vialetto-Van Laere syndrome (**González-Pérez et al., 2012**). These data suggest a role for ubiquilins in motor neuron diseases. *UBQLN4*, like *UBQLN1* and *UBQLN2*, is widely expressed and shows substantial homology to human *UBQLN2* (**Marín, 2014**). To test if genetic variants of *UBQLN4* are involved in the etiology of ALS, we screened its 11 exons with primers covering coding regions and the exon-intron boundaries. We examined 267 familial ALS index cases and 411 sporadic ALS cases, and identified a variant in a familial ALS case (**Figure 1A**). This variant, c.269A>C, was located in exon 3, leading to the change of aspartate to alanine, p.D90A at the protein level (**Figure 1C**). The female patient (III3) with this $UBQLN4^{D90A}$ variant had an age of disease onset at 55 years, with disease duration of two years. Her mother (II3), maternal aunt (II2) and a maternal cousin (III1) also developed ALS and died of respiratory failure two to five years following disease onset. The variant was neither present in our 332 in-house controls, nor in any SNP databases with a total of >15,000 sequenced alleles nor in the Exome Aggregation Consortium (ExAC) database in a total of 60,706 unrelated individuals. The amino acid D90 is adjacent to the

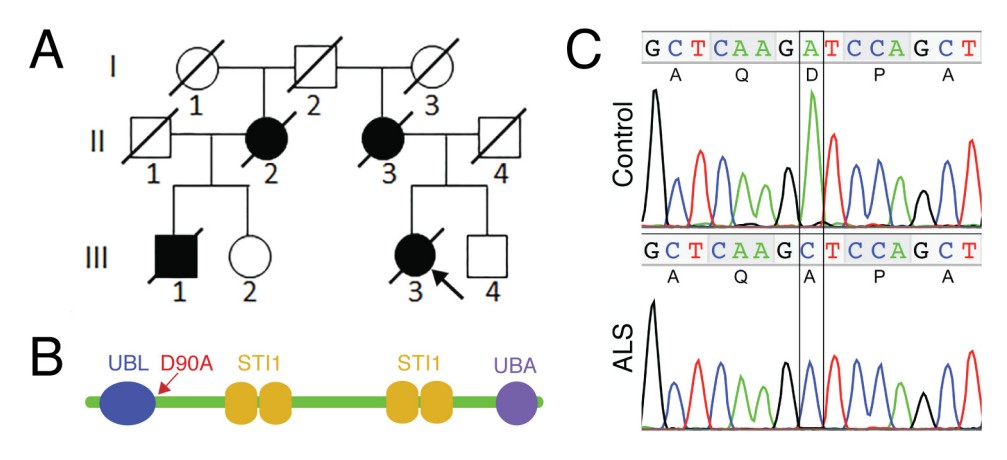

**Figure 1.** The *UBQLN4* c.269A>C (p.D90A) variant identified in a familial ALS case. (**A**) Pedigree of a family with ALS. The proband (III3, arrow) had disease onset at 55 years of age, with disease duration of 22 months. Her mother (II3) died of ALS at 62 years of age without clear information regarding disease onset. Her maternal grandfather (I2) died in a traffic accident without any known neurological problems. Her maternal aunt (II2) developed ALS with disease onset at 51 years of age, and disease duration of 36 months. Her cousin (III1) developed ALS at 56 years of age and died five years later. (**B**) Predicted structural and functional domains of UBQLN4 with an arrow indicating the position of the mutation site. Domains include a UBL: ubiquitin-like domain, aa 13–83; four STI1 heat-shock-chaperonin-binding motifs, aa 192–229, 230–261, 393–440 and 444–476; and a UBA: ubiquitin-associated domain, aa 558–597. (**C**) Sequencing chromatograms of *UBQLN4* wild-type allele in control and mutant allele in the patient with ALS. An adenine to cytosine substitution is present in the ALS patient, resulting in the change from aspartate to alanine at the ninetieth amino acid, D90A.

UBL domain of UBQLN4 (*Figure 1B*) and is highly conserved during evolution, suggesting its importance in structural and functional properties of the protein.

To assess the effects of the ALS-associated variant, we expressed wild-type or disease-associated UBQLN4 in cultured mouse spinal motor neurons (*Figure 2A*, *Figure 2—figure supplement 1*). UBQLN4$^{D90A}$-expressing cells showed a significant increase in the total number of neurites as compared to cells expressing wild-type UBQLN4, or non-transfected cells (*Figure 2A,C*). Importantly, these results were validated in vivo in zebrafish. When mRNAs encoding *UBQLN4-WT* or *UBQLN4$^{D90A}$* were injected into zebrafish embryos, we observed abnormal motor axon branching in *UBQLN4$^{D90A}$* but not *UBQLN4-WT* or uninjected embryos (*Figure 2B,D*). Both *UBQLN4-WT* and *UBQLN4$^{D90A}$*-injected fish embryos otherwise developed normally and showed neither gross morphological abnormalities nor significant changes in motor axon length, suggesting specificity of the motor axon branching phenotype. In all experiments, expression levels of UBQLN4-WT and UBQLN4$^{D90A}$ were comparable (*Figure 2—figure supplement 2A–C*). These results indicate that the ALS-associated *UBQLN4* variant interferes with normal motor axon morphogenesis in culture and in vivo.

## Expression of UBQLN4$^{D90A}$ leads to reduced proteasomal efficiency and accumulation of beta-catenin

Given the role ubiquilins play in the UPS, we sought to determine if UBQLN4$^{D90A}$ affects proteasome-mediated degradation. We used the Ub$^{G76V}$-GFP fusion protein as a reporter for UPS function (*Dantuma et al., 2000*). The G76V substitution produces an uncleavable ubiquitin moiety that acts as a proteasome degradation signal, thereby targeting GFP for proteasomal degradation. Rates of UPS-dependent protein turnover can therefore be monitored by GFP protein level. We transfected NSC-34 cells, a motor neuron-derived cell line, with Ub$^{G76V}$-GFP, Ub$^{G76V}$-GFP + UBQLN4-WT, or Ub$^{G76V}$-GFP + UBQLN4$^{D90A}$ and compared GFP levels. UBQLN4$^{D90A}$ expression resulted in reduced protein turnover, as indicated by significantly greater GFP signal compared to UBQLN4-WT or Ub$^{G76V}$-GFP alone (*Figure 3A,B*). We further confirmed this finding with cycloheximide protein

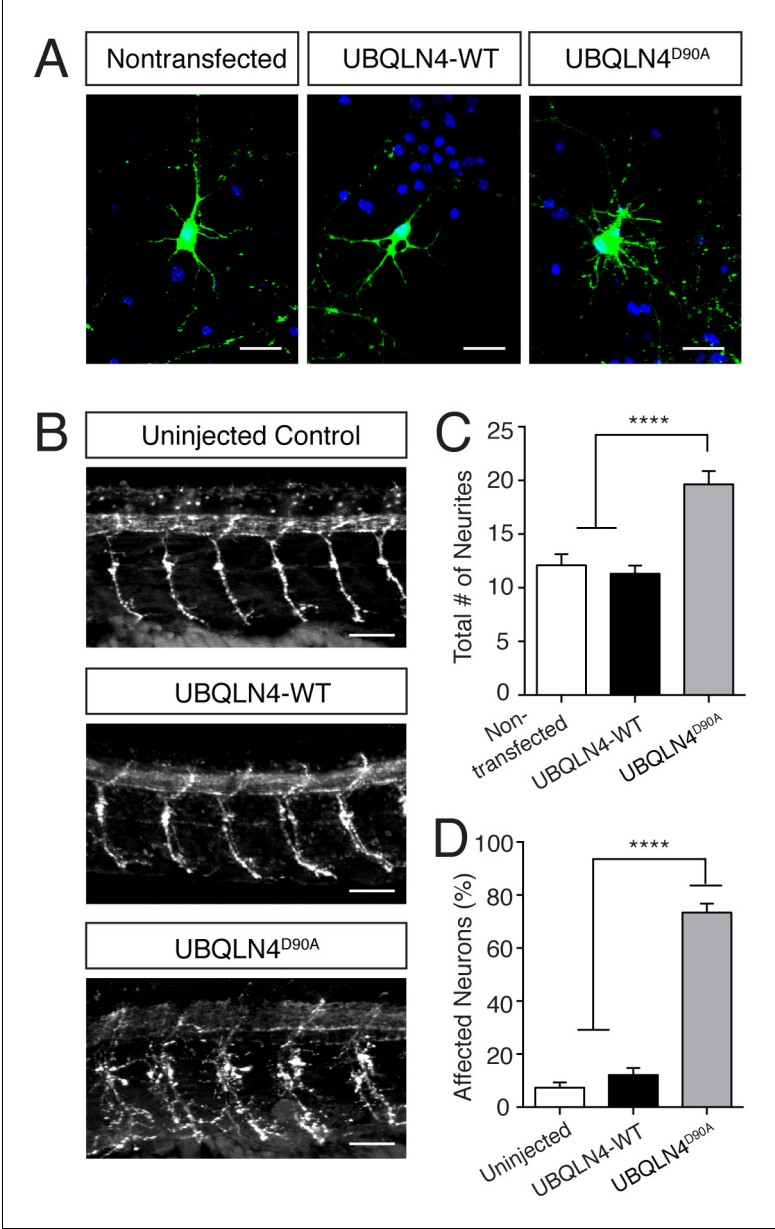

**Figure 2.** Expression of UBQLN4$^{D90A}$ results in motor axon branching abnormalities in vitro and in vivo. (**A**) Representative images of primary mouse spinal motor neurons transfected with pCAG-GFP alone, or co-transfected with UBQLN4-WT or UBQLN4$^{D90A}$. Scale bar: 20 µm. (**B**) Representative images of lateral whole-mount zebrafish spinal cords from uninjected, *UBQLN4-WT* mRNA, or *UBQLN4$^{D90A}$* mRNA injected embryos. Scale bar: 50 µm. (**C**) Quantification of total neurites in (A) revealed an increase in neurite number in UBQLN4$^{D90A}$ transfected neurons compared to pCAG-GFP-only and UBQLN4-WT transfected neurons (n = 30 cells per group, p<0.0001). Data are quantified from three independent experiments and are mean ± SEM. ****p<0.0001, one-way ANOVA with Bonferroni post-hoc test. (**D**) Quantification of percentage of motor axons with aberrant branching in (B) revealed an increase in the percentage of affected motor axons in *UBLQN4$^{D90A}$* injected zebrafish compared to both uninjected and *UBQLN4-WT* injected controls (n = 36 embryos per group, p<0.0001). The difference between uninjected and *UBQLN4-WT* injected fish was not significant (p=0.155). The average motor axon length was not significantly different among three groups (p=0.2034). Data are from three independent experiments and are mean ± SEM. ****p<0.0001, one-way ANOVA with Bonferroni post-hoc test.

The following figure supplements are available for figure 2:

**Figure supplement 1.** Cultured primary motor neurons express the motor neuron marker Islet1.

*Figure 2 continued on next page*

*Figure 2 continued*

**Figure supplement 2.** UBQLN4-WT and UBQLN4$^{D90A}$ are expressed at similar levels in primary mouse motor neurons and in zebrafish embryos.

stability assay (*Figure 3C,D*). Cells transfected with Ub$^{G76V}$-GFP alone or Ub$^{G76V}$-GFP + UBQLN4-WT showed a gradual reduction in GFP levels due to proteasomal degradation in the presence of protein synthesis inhibitor cycloheximide. Cells transfected with Ub$^{G76V}$-GFP + UBQLN4$^{D90A}$ showed much less GFP degradation, indicating impairment of proteasomal function by the ALS-associated *UBQLN4$^{D90A}$* variant (*Figure 3C,D*). Taken together, these results suggest that UBQLN4$^{D90A}$ impairs proteasomal function.

Recently, loss of the ubiquitin-like modifier-activating enzyme 1 (UBA1) was shown to cause motor axon abnormalities similar to what we observed. In the UBA1 study, the phenotypes were shown to result from heightened beta-catenin signaling, caused by loss of Uba1 and dysfunction of ubiquitination pathways (*Wishart et al., 2014*). Given our finding that UBQLN4$^{D90A}$ results in reduced proteasomal efficiency, we sought to determine if it also affects beta-catenin levels. We first compared beta-catenin levels in UBQLN4-WT, UBQLN4$^{D90A}$ and non-transfected NSC-34 cells (*Figure 3E*). There was a significant decrease in beta-catenin in UBQLN4-WT-expressing cells compared to non-transfected cells, suggesting a role for UBQLN4 in facilitating beta-catenin degradation. Moreover, we detected a significant increase of beta-catenin in UBQLN4$^{D90A}$-expressing cells compared to those expressing UBQLN4-WT, suggesting that the UBQLN4$^{D90A}$ mutation disrupts beta-catenin degradation (*Figure 3F*). Immunostaining in primary mouse spinal motor neurons validated these observations, and further confirmed nuclear beta-catenin accumulation in UBQLN4$^{D90A}$-expressing cells (*Figure 3G,H*).

## Inhibition of beta-catenin mitigates *UBQLN4$^{D90A}$* variant-induced motor axon morphogenic abnormalities

Given our findings that UBQLN4$^{D90A}$ expression results in reduced proteasomal efficiency and nuclear beta-catenin accumulation, we asked if inhibition of beta-catenin could mitigate motor axon morphogenesis defects characteristic of UBQLN4$^{D90A}$ expression. To address this possibility we utilized quercetin, which acts by restricting beta-catenin nuclear localization to reduce beta-catenin-dependent signaling (*Park et al., 2005*). Immunostaining for beta-catenin in mouse spinal motor neurons transfected with UBQLN4-WT or UBQLN4$^{D90A}$ revealed significantly greater nuclear accumulation of beta-catenin in UBQLN4$^{D90A}$-expressing cells as compared to UBQLN4-WT-expressing cells (*Figure 4A,B*). This accumulation was reduced by treatment with 0.1 µM quercetin, confirming the inhibitor's effectiveness. We next looked at quercetin's effects on axon morphogenesis in primary neurons and in zebrafish in the context of UBQLN4-WT or UBQLN4$^{D90A}$ expression. Indeed, the total number of neurites in 0.1 µM quercetin-treated UBQLN4$^{D90A}$-expressing neurons was rescued to wild-type levels (*Figure 4C,D*). Moreover, the percentage of motor axons with aberrant branching morphology in 50 µM quercetin-treated *UBQLN4$^{D90A}$*-expressing zebrafish embryos was significantly reduced as compared to vehicle-treated embryos (*Figure 4E and F*). Taken together, these findings suggest that motor axon morphogenesis phenotypes characteristic of UBQLN4$^{D90A}$ expression may result from beta-catenin accumulation, and that inhibition of beta-catenin is sufficient to mitigate UBQLN4$^{D90A}$ variant-induced phenotypes.

## Discussion

Here we provide the first evidence of *UBQLN4* involvement in ALS, and reveal the association of beta-catenin-dependent signaling with the disease variant-induced phenotypes. Taken with reports of contributions by *UBQLN1* and *UBQLN2* to neurodegeneration, our findings suggest prominent neuropathological involvement of the *UBQLN* gene family. In the present study, we identified abnormalities in spinal motor neuron morphogenesis in primary mouse neurons as well as a zebrafish model in vivo. Furthermore, we demonstrated that the novel ALS-associated *UBQLN4* variant impaired UPS function, leading to increased beta-catenin in UBQLN4$^{D90A}$-expressing cells. We also

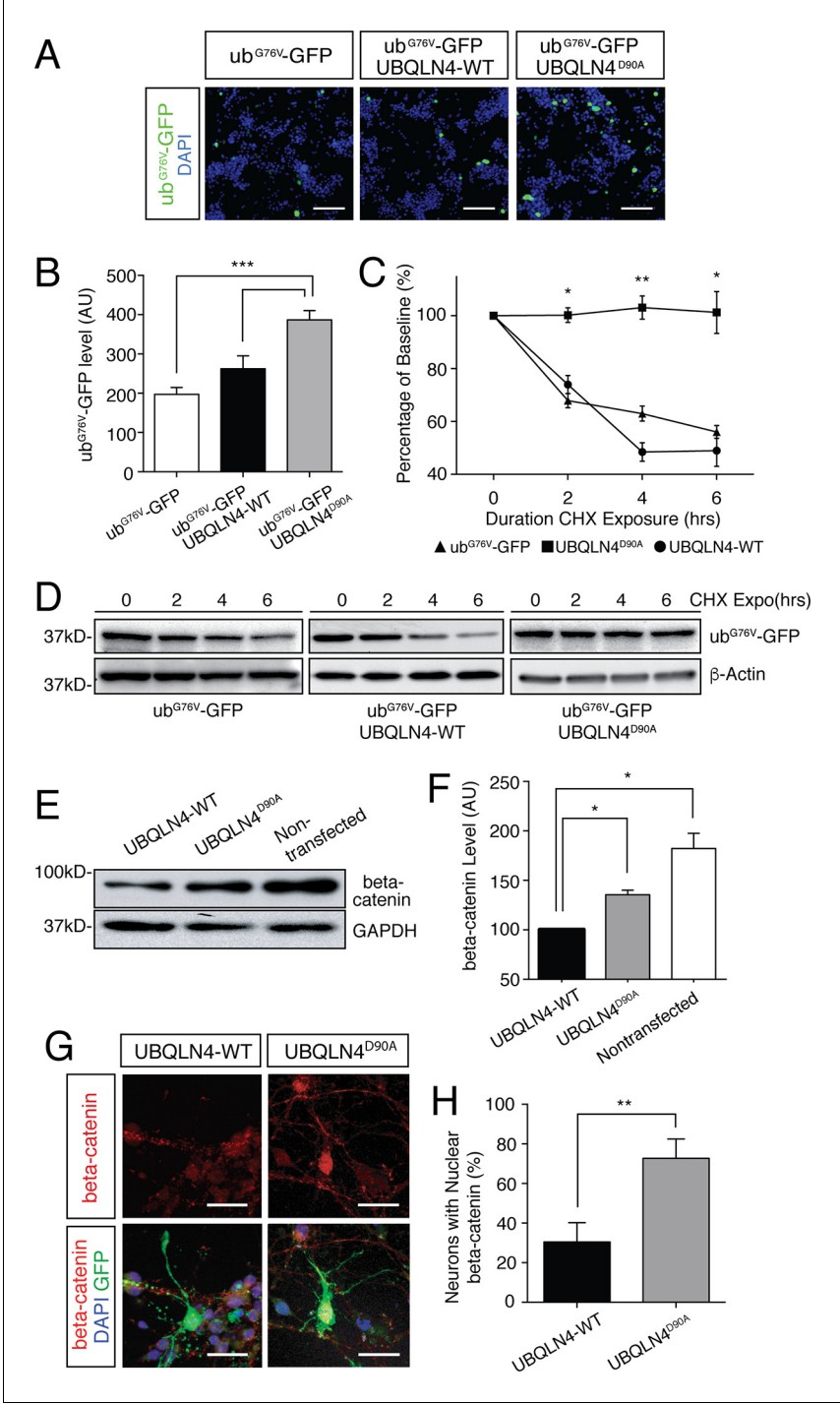

**Figure 3.** Expression of UBQLN4[D90A] impairs proteasomal degradation and results in beta-catenin accumulation. (**A**) Representative images of NSC-34 cells transfected with Ub[G67V]-GFP alone, or co-transfected with UBQLN4-WT or UBQLN4[D90A]. DAPI staining is shown in blue. Scale bar: 100 μm. (**B**) Quantification of GFP levels in (A) revealed reduced proteasomal turnover following UBQLN4[D90A] expression. GFP signal, normalized to DAPI, was greater in UBQLN4[D90A] transfected cells than in Ub[G67V]-GFP-only transfected cells (p=0.0008), or UBQLN4-WT transfected cells (p=0.0088). The difference in GFP level between Ub[G67V]-GFP -only and UBQLN4-WT transfected cells was not significant (p=0.1114). Results are from three independent experiments and are mean ± SEM. ***p<0.001, **p<0.01, one-way ANOVA with Bonferroni post-hoc test. (**C**) Cycloheximide protein stability assay. GFP protein stability is compared between Ub[G67V]-GFP-only, Ub[G67V]-GFP + UBQLN4-WT, and Ub[G67V]-GFP + UBQLN4[D90A] transfected NSC-34 cells treated with cycloheximide for 0, 2, 4, and 6 hr. Quantification of GFP level revealed

*Figure 3 continued on next page*

*Figure 3 continued*

impeded protein turnover in UBQLN4$^{D90A}$ transfected cells. GFP level of Ub$^{G67V}$-GFP + UBQLN4$^{D90A}$ transfected cells was significantly greater than that of Ub$^{G67V}$-GFP + UBQLN4-WT transfected cells at 2, 4, and 6 hr with cycloheximide treatment (p=0.024 (2 hr), p=0.0038 (4 hr), and p=0.036 (6 hr)). GFP level of Ub$^{G67V}$-GFP + UBQLN4$^{D90A}$ transfected cells was significantly greater than that of Ub$^{G67V}$-GFP-only transfected cells (p=0.0068 (2 hr), p=0.0093 (4 hours), and p=0.032 (6 hr)). Results are from three independent experiments and are mean ± SEM. **p<0.01, *p<0.05, one-way ANOVA with Bonferroni post-hoc test. (D) Representative Western blot of the cycloheximide protein stability assay. Actin serves as a loading control. (E) Western blot of beta-catenin levels from UBQLN4-WT, UBQLN4$^{D90A}$ and non-transfected NSC-34 cells. GAPDH Western blot indicates equal protein loading. (F) Quantification of beta-catenin signal in (D) indicated greater beta-catenin levels in UBQLN4$^{D90A}$ transfected and non-transfected cells as compared to UBQLN4-WT transfected cells (p=0.0174 and p=0.0326, respectively). Results are from three independent experiments and are mean ± SEM. *p<0.05, one-way ANOVA with Bonferroni post-hoc test. (G) Representative images of primary mouse neurons transfected with pCAG-GFP and UBQLN4-WT or UBQLN4$^{D90A}$, stained for beta-catenin. Scale bar: 20 μm. (G) Quantification of beta-catenin localization in (F) revealed increased nuclear localization of beta-catenin in UBQLN4$^{D90A}$ transfected cells as compared to UBQLN4-WT transfected cells (*n* = 22 or more cells per group, p=0.0038). Data are from three independent experiments and are mean ± SEM. **p<0.01, two-tailed Student's t-test.

showed that inhibition of beta-catenin was sufficient to mitigate these phenotypes, suggesting a role for beta-catenin signaling in regulating these physiological and pathological functions.

Ubiquilins are characterized by a ubiquitin-like UBL domain and a ubiquitin-associated UBA domain that interact with proteasomes and polyubiquitinated substrates, respectively, to facilitate proteasomal degradation (*Ko et al., 2004*). Because the D90A mutation lies near the UBL domain, the UBQLN4$^{D90A}$ variant may compromise interaction with proteasomes, consistent with our finding of proteasomal impairment in UBQLN4$^{D90A}$-expressing neurons. Beta-catenin-dependent signaling has been shown to play a critical role in regulating multiple aspects of neuronal development, including neurite outgrowth (*Votin et al., 2005*), axon guidance (*Avilés and Stoeckli, 2016*; *Maro et al., 2009*), and target innervation (*Salinas and Zou, 2008*; *Wu et al., 2012*). Therefore UBQLN4$^{D90A}$ expression, which compromises UPS function and causes beta-catenin accumulation, may affect motor neuron development through dysregulated beta-catenin-dependent signaling. Early neurodevelopmental processes affect mature neuron functions; motor axon morphogenesis is essential for action potential transmission and target innervation. Defects in these processes may therefore confer functional vulnerability at later stages, rendering motor neurons susceptible to degeneration in ALS. This is consistent with reports of aberrant motor axon morphology associated with ALS-related genes including TDP-43 (*Kabashi et al., 2010*), SOD1 (*Clark et al., 2016*; *Lemmens et al., 2007*; *Ramesh et al., 2010*) and C9orf72 (*Burguete et al., 2015*; *Ciura et al., 2013*).

Our observation that UBQLN4$^{D90A}$ expression impairs proteasome function (*Figure 3A–D*) suggests that the variant acts in a dominant-negative manner. However, our finding that beta-catenin level was still reduced in UBQLN4$^{D90A}$ expressing cells compared to nontransfected cells (*Figure 3E, F*) suggests that the variant results in a partial loss-of-function. This may be because in the short term UBQLN4$^{D90A}$ functions with reduced efficiency, but still facilitates degradation of substrates. Overtime, reduced UBQLN4 functional efficiency may overload the proteasome pathway; therefore, an initial partial loss of function can eventually lead to a dominant negative gain of function outcome. This notion is consistent with the progressive nature of ALS.

It is notable that our findings link *UBQLN4* with beta-catenin in the context of ALS, as recent work suggested a similar role for beta-catenin signaling in the pediatric motor neuron disease Spinal Muscular Atrophy (SMA) (*Wishart et al., 2014*). Wishart et al. reported similar morphological phenotypes in UBA1-deficient zebrafish motor neurons in which beta-catenin expression was heightened, and showed a dose-dependent rescue effect by quercetin. The degradation of beta-catenin is known to be carried out through the ubiquitin proteasome pathway (*Aberle et al., 1997*). Disruption of *UBQLN4* or UBA1 function, which are both involved in the UPS pathway, could therefore lead to beta-catenin accumulation and aberrant signaling. Given the wide-ranging and critical roles in neuronal development and function played by beta-catenin, its upregulation in ALS and SMA may understandably have dire consequences for the development, function, and survival of affected motor neurons. Collectively these studies suggest heightened beta-catenin activity as a common

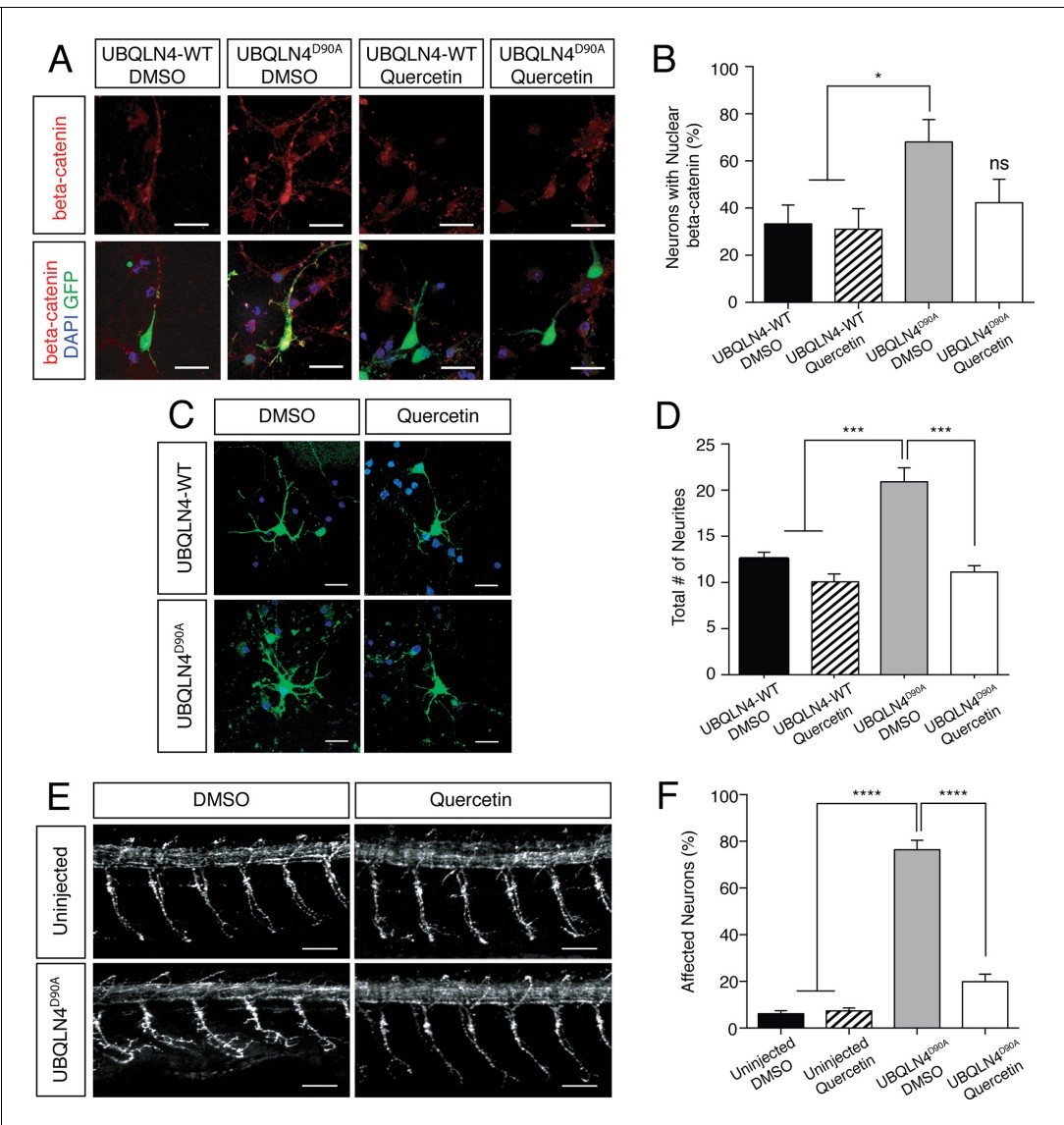

**Figure 4.** UBQLN4$^{D90A}$-induced phenotypes are rescued by beta-catenin inhibition. (**A**) Representative images of primary mouse neurons transfected with pCAG-GFP and UBQLN4-WT or UBQLN4$^{D90A}$, treated with 0.1 μM quercetin or DMSO. Cells are stained for beta-catenin. Scale bar: 20 μm. (**B**) Quantification of beta-catenin localization in (**A**) revealed a rescue effect of quercetin on increased nuclear localization of beta-catenin caused by UBQLN4$^{D90A}$. UBQLN4$^{D90A}$ transfected cells showed a dramatic increase of nuclear beta-catenin localization compared to UBQLN4-WT transfected cells (n = 25 or more cells per group, p<0.05). The increase was rescued by the application of quercetin (n = 25 or more cells per group, p=0.48). Data are from three independent experiments and are mean ± SEM. *p<0.05, one-way ANOVA with Bonferroni post-hoc test. (**C**) Representative images of primary mouse spinal cord neurons transfected with pCAG-GFP and UBQLN4-WT or UBQLN4$^{D90A}$, treated with 0.1 μM quercetin or DMSO. Scale bar: 20 μm. (**D**) Quantification of total neurite numbers in (**C**) revealed a rescue effect of quercetin on increased neurite number in UBQLN4$^{D90A}$ transfected cells. The number of neurites present in UBQLN4$^{D90A}$ transfected cells was significantly greater than that in UBQLN4-WT transfected cells (n = 30 cells per group, p<0.0001). Quercetin treatment rescued the increased number of neurites induced by UBQLN4$^{D90A}$ transfection (n = 30 cells per group, p<0.0001). Data are from three independent experiments and are mean ± SEM. ****p<0.0001, one-way ANOVA with Bonferroni post-hoc test. (**E**) Representative images of lateral whole-mount zebrafish spinal cord from uninjected controls or *UBQLN4$^{D90A}$* mRNA injected embryos, treated with DMSO or 50 μM quercetin. Scale bar: 50 μm. (**F**) Quantification of the percentage of motor axons with aberrant branching in (**E**) revealed a rescue effect of quercetin on *UBQLN4$^{D90A}$* injected embryos. *UBQLN4$^{D90A}$* injected embryos showed a significantly greater percentage of affected motor axons compared to uninjected controls (n = 60 embryos per group, p<0.0001). Quercetin treatment rescued aberrant motor axon branching in *UBQLN4$^{D90A}$* injected embryos (n = 60 embryos per group, p<0.0001). Data are from three independent experiments and are mean ± SEM. ****p<0.0001, one-way ANOVA with Bonferroni post-hoc test.

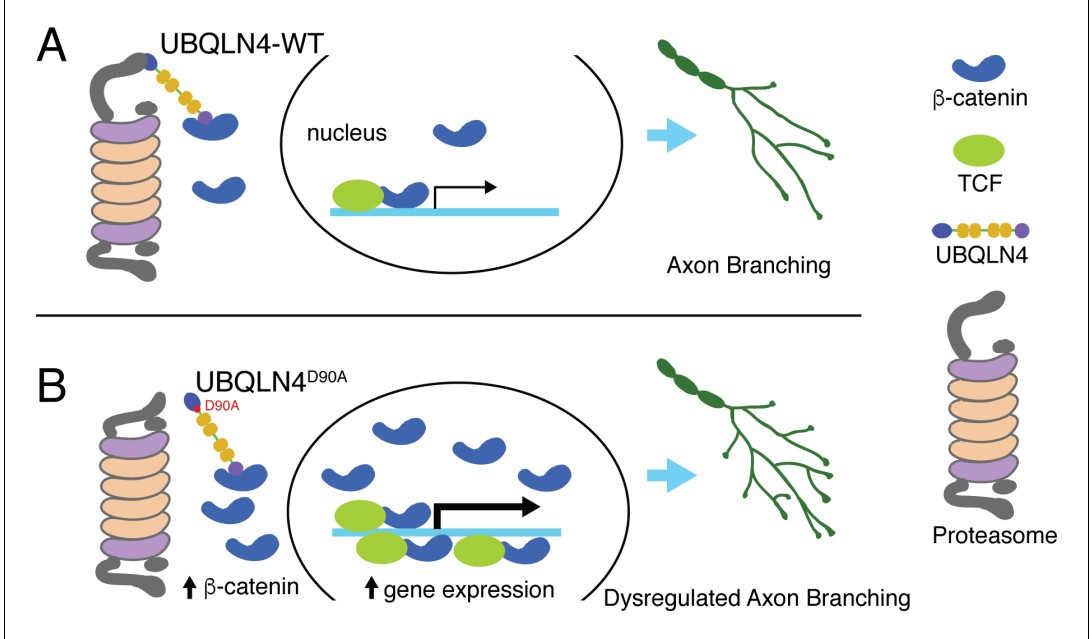

**Figure 5.** Schematic model illustrating proposed roles for wild-type (A) and ALS-associated *UBQLN4^{D90A}* (B) in motor axon morphogenesis. (**A**) Wild-type *UBQLN4* associates with beta-catenin through its UBA domain, and with the proteasome through its UBL domain. These interactions allow for the degradation of beta-catenin, which in turn modulates gene expression to control motor axon morphogenesis. (**B**) The ALS-associated *UBQLN4^{D90A}* variant is deficient in mediating proteasomal degradation of beta-catenin, leading to its accumulation and excessive induction of gene expression. Hyperactivation of beta-catenin-controlled genes dysregulates axon morphogenesis, causing aberrant axon branching in motor neurons.

mechanism between the adult-onset motor neuron disease ALS and the pediatric motor neuron disease SMA.

Taken together, we revealed a novel role for *UBQLN4* in regulating motor axon morphogenesis through the UPS. Dysregulation of this function by the ALS-associated *UBQLN4^{D90A}* variant leads to compromised proteasome function and beta-catenin accumulation, conferring abnormalities in motor axon morphogenesis (*Figure 5*), and contributing to motor neuron degeneration in ALS. Further exploration of the underlying mechanism may provide new insights for understanding ALS pathogenesis and for therapeutic development.

## Materials and methods

### ALS disease-associated variant sequencing

This study has been approved by the Northwestern University Institutional Review Board. Blood samples were collected after obtaining written informed consent. Eleven sets of primers (*Supplementary file 1*) were synthesized for PCR amplification of human *UBQLN4* exons and Sanger sequencing using the EQ 8000 Genetic Analysis System (*Deng et al., 2011*). A total of 267 familial ALS index cases and 411 sporadic ALS cases were sequenced. The familial case in which the *UBQLN4* variant was identified was compared to numerous control large-scale reference datasets to validate the variant: these included 332 in-house controls, SNP databases with a total of >15,000 sequenced alleles, and the Exome Aggregation Consortium (ExAC) database with a total of 60,706 unrelated individuals (*Lek et al., 2016*).

### Animals

This study has been approved by the Institutional Animal Care and Use Committee of the Lurie Children's Hospital of Chicago. All studies were conducted in accordance with the US Public Health Service's Policy on Humane Care and Use of Laboratory Animals. Wild-type AB zebrafish (RRID:

ZIRC_ZL1) were obtained from the Zebrafish International Resource Center (Eugene, OR) and maintained in standard conditions. Timed-pregnant wild-type CD1 mice (RRID:IMSR_CRL:086) were obtained from Charles River Laboratories (Chicago, IL).

## Constructs and cloning

A human UBQLN4 cDNA clone was obtained from GE Dharmacon (Clone ID: 6183942, Accession #BU149502). The coding region was excised and 5'Fse1 and 3'Asc1 sites were inserted via PCR (*UBQLN4* FseFwd 5' GATC GGC CGG CCT ACC ATG GCG GAG CCG AGC GGG GCC GAG 3'; *UBQLN4* AscRev 5' GAT CGG CGC GCC TTA GGA GAG CTG GGA GCC CAG CAG 3'). The product was ligated into a pCS2-Flag vector, and site-directed mutagenesis (Q5 kit, NEB) was performed to convert the 2747 adenine to cytosine (*UBQLN4* Q5 Fwd 5' AAG GCT CAA GcT CCA GCT GCT G 3'; *UBQLN4* Q5 Rev 5' CTG AGG GGT CTT GAT GAC 3'). Both wild-type and mutant constructs were verified by sequencing. Ub$^{G76V}$-GFP was a gift from Nico Dantuma (Addgene plasmid #11941).

## mRNA microinjection in zebrafish

Capped mRNAs were generated from linearized UBQLN4 wild-type and mutant constructs via in vitro transcription using the mMessage mMachine SP6 Transcription Kit (Ambion). Zebrafish embryos were injected with capped mRNA (200–300 pg target) at the single-cell stage and grown until 32 hr post-fertilization. Embryos were treated with quercetin (50 µM in 0.25% DMSO in embryo medium) or vehicle control from six hours post-fertilization until fixation.

## Cell culture and transfection

Primary mouse spinal cord motor neurons were isolated, dissociated, and cultured as described previously (*Miller et al., 2015*). The NSC-34 motor neuron cell line (RRID:CVCL_D356) was provided by Dr. Neil Cashman (University of British Columbia, Vancouver, Canada; [*Cashman et al., 1992*]). The NSC-34 line is not included in the International Cell Line Authentication Committee's Database of Cross-Contaminated or Misidentified Cell Lines, and is negative for mycoplasma. Primary and NSC-34 cells were transfected using Lipofectamine 2000 (Life Technologies) according to manufacturer's instructions. Cultures were fixed for imaging or lysed for Western blot analysis 48 hr post-transfection. For cycloheximide protein stability assay, cells were treated for 0, 2, 4, or 6 hr with 100 µg/ml cycloheximide following Ub$^{G76V}$-GFP transfection. For quercetin treatment, a working concentration of 0.1 µM was used.

## Immunoblotting and staining

Western blotting was performed as described (*Miller et al., 2015*) with the following antibodies and dilutions: mouse anti-beta-catenin (BD-Biosciences #610153, RRID:AB_397554; 1:500), goat anti-beta-actin (Santa Cruz #1616, RRID:AB_630836; 1:1000), rabbit anti-GAPDH (Santa Cruz #25778, RRID:AB_10167668; 1:1000), mouse anti-GFP (Affymetrix #14-6674-80, RRID:AB_2572899; 1:300), and mouse anti-Flag M2 (Sigma #F3165, RRID:AB_259529, 1:1000). Whole-mount immunostaining of zebrafish embryos was performed as follows: embryos were fixed overnight in 4% PFA at 4°C. The tissue was permeabilized with proteinase K (10 µg/ml) for 40 min and blocked with 5% BSA and 1% goat serum in PBS for one hour. Embryos were incubated in primary antibody, mouse anti-znp1 (DSHB #znp - 1, RRID:AB_2315626; 1:500), in blocking solution overnight at 4°C. Goat-anti-mouse Cy3 secondary antibody (Jackson ImmunoResearch, 1:250) diluted in blocking solution was applied for one hour at room temperature. Immunostaining of cultured mouse neurons was performed as described (*Miller et al., 2015*) with the following antibodies and dilutions: mouse anti-DDDDK tag M2 (Abcam #ab45766, RRID:AB_731867; 1:1500), mouse anti-Isl1 (DSHB #40.2D6, RRID:AB_528315; 1:60), and mouse anti-beta-catenin (BD-Biosciences #610153, RRID:AB_397554; 1:300). All images were acquired with the Zeiss 510 Meta Laser Scanning Microscope.

## Acknowledgements

This work was supported by NIH grants NS094564, AG043970 and grants from The Hartwell Foundation and Whitehall Foundation (YCM); NS078504, NS078504, Les Turner ALS Foundation, the Foglia Family Fund for ALS Research, Les Turner ALS Foundation/Herbert C Wenske Foundation

Professorship, Vena E Schaff ALS Research Fund, and SP Foundation (TS). YCM is Ann Marie and Francis Klocke MD Research Scholar supported by the Joseph and Bessie Feinberg Foundation.

## Additional information

### Funding

| Funder | Grant reference number | Author |
| --- | --- | --- |
| National Institute of Neurological Disorders and Stroke | NS094564 | Yongchao C Ma |
| National Institute on Aging | AG043970 | Yongchao C Ma |
| The Hartwell Foundation | | Yongchao C Ma |
| Whitehall Foundation | | Yongchao C Ma |
| National Institute of Neurological Disorders and Stroke | NS078504 | Teepu Siddique |
| Les Turner ALS Foundation | | Teepu Siddique |

The funders had no role in study design, data collection and interpretation, or the decision to submit the work for publication.

### Author contributions

BME, Data curation, Formal analysis, Investigation, Writing—original draft, Writing—review and editing; JY, NM, Investigation; NM, Performed mouse spinal cord dissection; H-XD, TS, Data curation, Formal analysis, Writing—review and editing; YCM, Conceptualization, Data curation, Formal analysis, Writing—original draft, Writing—review and editing

### Author ORCIDs

Yongchao C Ma, http://orcid.org/0000-0002-2469-4356

### Ethics

Human subjects: The use of human subjects in this study has been approved by the Northwestern University Institutional Review Board (IRB). Informed consent was obtained from all subjects.

Animal experimentation: All animal use in this study has been approved by the Institutional Animal Care and Use Committee (IACUC) of the Lurie Children's Hospital of Chicago (protocols 14-012 and 15-006). All studies were conducted in accordance with the US Public Health Service's Policy on Humane Care and Use of Laboratory Animals.

## Additional files

### Supplementary files

• Supplementary file 1. Human *UBQLN4* sequencing primers.

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
