## [Decision Letter]

Thank you for submitting your article "A Novel ALS-Associated Variant in *UBQLN4* Regulates Motor Axon Morphogenesis" for consideration by *eLife*. Your article has been favorably evaluated by Marianne Bronner (Senior Editor) and three reviewers, one of whom is a member of our Board of Reviewing Editors. The following individual involved in review of your submission has agreed to reveal his identity: Thomas H Gillingwater (Reviewer #3).

The reviewers have discussed the reviews with one another and the Reviewing Editor has drafted this decision to help you prepare a revised submission.

Summary:

This manuscript from Edens and colleagues reports on a timely and potentially important set of experiments, detailing a novel variation in the *UBQLN4* gene associated with ALS. While this in itself may not be particularly novel (mutations in other ubiquilin genes have been linked to neurodegenerative conditions, including motor neuron disease), the subsequent demonstration of a mechanism of action is exciting and is likely to be of considerable interest to a broad readership. The authors have used a range of in vitro and in vivo approaches to show that the *UBQLN4^D90A^* variant; i) modulates motor axon stability/morphology, ii) affects proteasome-mediated degradation, leading to an abnormal accumulation of β-catenin, and iii) modulates motor axon stability via a β-catenin-dependent pathway, with inhibition of β-catenin ameliorating motor neuron defects. The short paper is, in general, well written and the study has been well-designed and performed. Only a few issues need to be addressed.

Essential revisions:

1) For the in vitro data shown in Figure 2 (and Figure 3), the authors need to provide evidence that the cells they are quantifying are indeed motor neurons (e.g. co-label with motor neuron-specific markers). Were there any other cellular defects in these motor neurons alongside changes in neurite number (e.g. evidence for activation of cell death pathways and/or changes in the size of the neuronal soma)? Also, these data should be quantified in the same way the zebrafish axonal branching defects were quantified.

2) For both the in vitro and in vivo experiments in Figure 2, the authors need to provide data showing the level of overexpression achieved in their *UBQLN4-WT* and *UBQLN4^D90A^* cells/animals. Were the *UBQLN4^D90A^* animals otherwise healthy?

3) Example raw GFP images should be provided to accompany Figure 3.

4) The dose(s) of quercetin need to be clearly stated in the results and legend to Figure 4. Was there any evidence for a dose-dependent effect of quercetin? This is a particularly interesting finding of the manuscript, so elaboration with either experiments or discussion will be of strong interest to the readers.

5) Is the in vivo developmental branching phenotype observed in zebrafish in the current study similar to any other published zebrafish models of ALS, or does this appear to be a *UBQLN4^D90A^* specific phenotype? The similarity to zebrafish models of spinal muscular atrophy is quite striking, but did the authors find any evidence for truncated motor axons, as well as aberrantly-branched axons (c.f. McWhorter et al. 2003 J Cell Biol; Wishart et al. 2014 JCI)? If this can be addressed experimentally with further analysis of the data on hand, this should be added.

---

## [Author Response]

*Essential revisions:*

*1) For the in vitro data shown in Figure 2 (and Figure 3), the authors need to provide evidence that the cells they are quantifying are indeed motor neurons (e.g. co-label with motor neuron-specific markers). Were there any other cellular defects in these motor neurons alongside changes in neurite number (e.g. evidence for activation of cell death pathways and/or changes in the size of the neuronal soma)? Also, these data should be quantified in the same way the zebrafish axonal branching defects were quantified.*

To demonstrate that the cells we quantified are motor neurons, we now provide evidence in supplemental figure (Figure 2—figure supplement 1). Using immunofluorescence staining with motor neuron marker Isl1, we show in Figure 2—figure supplement 1 that the GFP-positive cells that we quantify are indeed motor neurons.

We did not find any significant changes in the size of the neuronal soma, activation of cell death pathways, or other features of *UBQLN4^D90A^*-expressing neurons.

Regarding the quantification of axon branching defects, we understand the reviewer’s concern. We present our data this way because essentially all mouse motor neurons expressing the ALS-associated *UBQLN4^D90A^* variant are defective in axon morphogenesis. Quantifying changes in total number of neurites represents a rigorous and objective way to present the results from mouse motor neurons. We request to maintain this way of reporting.

*2) For both the in vitro and in vivo experiments in Figure 2, the authors need to provide data showing the level of overexpression achieved in their UBQLN4-WT and UBQLN4^D90A^ cells/animals. Were the UBQLN4^D90A^ animals otherwise healthy?*

We have included anti-Flag immunofluorescence staining of primary mouse motor neurons transfected with mutant and WT *UBQLN4* plasmids, and anti-Flag Western blot analysis of zebrafish embryos injected with mutant and WT *UBQLN4* mRNA. In both experiments, levels of *UBQLN4*/Flag expression are comparable between different samples, and negative controls (nontransfected cells or uninjected fish) are indeed negative. These results have been included in Figure 2—figure supplement 2.

Regarding the second question, we did not note any gross morphological or general health abnormalities in the *UBQLN4^D90A^* expressing fish, suggesting that the motor axon phenotype is rather specific. We’ve added these observations to the revised manuscript.

*3) Example raw GFP images should be provided to accompany Figure 3.*

*4) The dose(s) of quercetin need to be clearly stated in the results and legend to Figure 4. Was there any evidence for a dose-dependent effect of quercetin? This is a particularly interesting finding of the manuscript, so elaboration with either experiments or discussion will be of strong interest to the readers.*

The dosage of quercetin has been added to the results and figure legend, in addition to being noted in the Methods section of the revised manuscript. While we did not perform a dose response experiment in the present study, we referenced Wishart et al. for the optimal quercetin dosage in zebrafish and multiple literature sources for the optimal dosage in primary neurons. We further elaborated on this point in the revised Discussion.

*5) Is the in vivo developmental branching phenotype observed in zebrafish in the current study similar to any other published zebrafish models of ALS, or does this appear to be a UBQLN4^D90A^ specific phenotype? The similarity to zebrafish models of spinal muscular atrophy is quite striking, but did the authors find any evidence for truncated motor axons, as well as aberrantly-branched axons (c.f. McWhorter et al. 2003 J Cell Biol; Wishart et al. 2014 JCI)? If this can be addressed experimentally with further analysis of the data on hand, this should be added.*

The branching phenotype we observed seems to be a common characteristic shared by other zebrafish models expressing ALS-associated genes, including *TDP-43* (Kabashi et al., 2010), *SOD1* (Clark et al., 2016; Lemmens et al., 2007; Ramesh et al., 2010) and *C9orf72* (Burguete et al., 2015; Ciura et al., 2013). We have cited these papers in the Discussion of our original manuscript.

In response to the reviewers’ second question, we have measured motor axon lengths from mutant and WT mRNA injected zebrafish embryos, and nontransfected embryos. Though we did observe truncated axons in a few mutant *UBQLN4*-injected samples, statistically the motor axon length was not different in the mutant group compared to the other two control groups. This further analysis has been added to the revised manuscript, as this finding may be of interest to readers.